# MicroRNAs as Epigenetic Determinants of Treatment Response and Potential Therapeutic Targets in Prostate Cancer

**DOI:** 10.3390/cancers13102380

**Published:** 2021-05-14

**Authors:** Valentina Doldi, Rihan El Bezawy, Nadia Zaffaroni

**Affiliations:** Molecular Pharmacology Unit, Department of Applied Research and Technological Development, Fondazione IRCCS Istituto Nazionale dei Tumori, 20133 Milan, Italy; Valentina.doldi@istitutotumori.mi.it (V.D.); Rihan.elbezawy@istitutotumori.mi.it (R.E.B.)

**Keywords:** microRNA, epigenetics, prostate cancer, therapy

## Abstract

**Simple Summary:**

Prostate cancer is one of the most common and lethal tumors in men worldwide. Due to the high heterogeneity of the disease, the optimal treatment choice for individual patients is very challenging, thus leading to treatment failure caused by poor responsiveness and/or tumor recurrence. Therefore, the identification of novel actionable targets involved in determining tumor treatment response is essential for developing patient-tailored therapeutic strategies. In this scenario, microRNAs, small endogenous RNA molecules able to epigenetically regulate cellular processes that are widely deregulated in cancer, have been proposed as potential treatment response modulators in many tumor types, including prostate cancer. In this review, we provide an overview on the main microRNAs involved in prostate cancer response to radiation and drug therapy, describing the mechanisms by which they concur to determine disease response, and illustrate whether they can be considered novel therapeutic targets/tools for improving treatment response in prostate cancer patients.

**Abstract:**

Prostate cancer (PCa) is the second most common tumor in men worldwide, and the fifth leading cause of male cancer-related deaths in western countries. PC is a very heterogeneous disease, meaning that optimal clinical management of individual patients is challenging. Depending on disease grade and stage, patients can be followed in active surveillance protocols or undergo surgery, radiotherapy, hormonal therapy, and chemotherapy. Although therapeutic advancements exist in both radiatiotherapy and chemotherapy, in a considerable proportion of patients, the treatment remains unsuccessful, mainly due to tumor poor responsiveness and/or recurrence and metastasis. microRNAs (miRNAs), small noncoding RNAs that epigenetically regulate gene expression, are essential actors in multiple tumor-related processes, including apoptosis, cell growth and proliferation, autophagy, epithelial-to-mesenchymal transition, invasion, and metastasis. Given that these processes are deeply involved in cell response to anti-cancer treatments, miRNAs have been considered as key determinants of tumor treatment response. In this review, we provide an overview on main PCa-related miRNAs and describe the biological mechanisms by which specific miRNAs concur to determine PCa response to radiation and drug therapy. Additionally, we illustrate whether miRNAs can be considered novel therapeutic targets or tools on the basis of the consequences of their expression modulation in PCa experimental models.

## 1. Introduction

### 1.1. Prostate Cancer Management

Prostate cancer (PCa) is the second most common non-cutaneous cancer affecting men. Although in recent years there has been a decrease in mortality due to earlier detection and advances in treatment, PCa represents the fifth-leading cause of cancer-related death in males [1]. On the basis of the Gleason Score, PSA levels, and clinical stage, PCa has been traditionally stratified as a low-, intermediate-, and high-risk disease [2]. As for many tumors, the determination of the risk is pivotal for guiding clinical management decisions. In PCa, the risk ascription is strictly related to an accurate PCa needle biopsy, which should reflect comprehensively the complexity of the tumor and provide a precise characterization of the disease. However, due to the multifocal nature of the disease, this diagnostic procedure misses a significant fraction of PCa foci, causing a high heterogeneity of PCa patient outcomes within each risk group stratification [3].

Depending on the clinical state, various management options are available for those who receive a PCa diagnosis, including conservative approach, radical treatments, hormone-based therapy, and chemotherapy for advanced disease [2,4]. Clinically, localized PCa can range from those displaying a low malignant potential (low-risk disease) to those that are curable with local radical prostatectomy or definitive radiotherapy (intermediate-risk disease). As regards low-risk localized PCa, the active surveillance conservative approach is considered a valuable alternative to radical treatments [5]. AS protocols are characterized by a strict monitoring of the disease progression, which involves serial PSA evaluation, sequential prostate biopsy, and physical examinations as monitoring of the disease stage with the final intent for cure by using radical intervention in those patients who carry clinically significant disease [6].

#### 1.1.1. Radical Prostatectomy and Radiotherapy

When a therapeutic intervention is needed, surgery and radiotherapy (RT) are the standard treatment approaches.

PCa surgery, referred to as radical prostatectomy, consists of the complete removal of the prostate gland, the seminal vesicles, and the surrounding lymph nodes [2]. Although it represents a gold standard treatment for PCa, radical prostatectomy is often accompanied by several adverse events that negatively impact patients’ quality of life. Interestingly, first-line RT ensures a therapeutic success equivalent to that of prostatectomy, yet displaying lower toxicity. For this reason, both radical prostatectomy and RT are equally proposed as effective therapeutic interventions, and the choice between the two treatment options is concerted between the patient and the physician [7].

Intensity-modulated external beam radiation therapy (IMRT) with image-guided RT is the gold standard of external beam radiation therapy because it is associated with less toxicity than conformal radiotherapy (3D-CRT) [8]. Radiotherapy can also be administered through the use of permanent radioactive seeds implanted into the prostate (brachytherapy), representing an option for low-risk and favorable intermediate-risk PCa [2].

Moreover, RT can be used as an adjuvant treatment after prostatectomy in high-risk, locally advanced PCa characterized by an increased risk of local relapse due to extra-capsular tumor growth [2].

Alternative modalities have emerged as possible therapeutic options in clinically localized PCa, including cryotherapy, high-intensity focused ultrasounds, and focal therapy [2].

#### 1.1.2. Androgen Deprivation Therapy

A significant fraction of patients (between 27% and 53%) who undergo radical prostatectomy or RT develop a rising PSA level (PSA recurrence) [4]. Salvage RT (SRT) provides a possibility of cure for patients showing PSA recurrence after radical prostatectomy. The addition of androgen deprivation therapy (ADT) to SRT has been reported to improve patient outcome [9,10]. Salvage option for patients undergoing PSA recurrence following RT includes salvage radical prostatectomy and salvage cryoablation of the prostate [2].

Metastatic PCa treatment is based on ADT and obtained by chemical castration agents, such as luteinizing hormone-releasing hormone (LHRH) agents and antagonists [11,12].

Combined androgen blockade by using modern androgen receptor-targeted agents (enzalutamide, apalutamide) plus abiraterone acetate, a CYP17 inhibitor, significantly improves clinical outcome. Again, the addition of the chemotherapeutic agent docetaxel was found to improve patient survival [4].

A considerable percentage of patients became rapidly refractory to ADT-based regimens and gradually progresses toward castration-resistant PCa (CRPC) within 5 years [13]. The recognition that CRPC is not a hormone-independent entity but remains fostered by androgens has led to the use of the second generation of AR-directed agents, such as abiraterone acetate and enzalutamide, which are able to target the androgen dependence of CRPC, showing efficacy against metastatic CRPC (mCRPC) as first-line treatment [14,15]. Mechanistically, abiraterone acetate can induce ablation of androgens by irreversibly inhibiting the key enzyme CYP17A involved in androgen biosynthesis, while enzalutamide acts by inhibiting the androgen receptor (AR) signaling by competitively binding testosterone site on AR [15].

#### 1.1.3. Chemotherapy

The taxane docetaxel represents the standard first-line chemotherapy for mCRPC patients, providing a significant improvement in median survival compared to the previous standard of care [16]. Interestingly, clinical trials demonstrated that abiraterone acetate [17] and enzalutamide [15] can have significant outcomes for docetaxel-pretreated patients. The immunotherapeutic agent Sipuleucel-T, which showed a survival benefit in asymptomatic/minimally symptomatic mCRPC patients, also represents a therapeutic option for the advanced disease.

Concerning second-line treatments for mCRPC, the novel taxane cabazitaxel was shown to display a significant benefit in docetaxel-resistant PCa in terms of survival and objective response [14]. Interestingly, recent trials demonstrated that abiraterone acetate can significantly improve outcomes of docetaxel pretreated patients [13,18].

Finally, the α-emitter Radium-233 is the only bone-specific drug associated with a survival benefit in mCRPC patients who failed or were unfit for docetaxel [19].

However, independently from the therapeutic strategy or the treatment timing and sequencing, a considerable percentage of patients develop refractory disease. In particular, after treatment with AR pathway inhibitors, such as enzalutamide and abiraterone, PCa cells can undergo transdifferentiation, leading to the emergence of neuroendocrine PCa (NEPC), which is characterized by AR signaling independence and low expression of AR, PSA, and PSMA. NEPC patients have a severe prognosis, which is attributed in part to the lack of effective therapeutic strategies [20].

To date, overcoming therapeutic resistance remains the main challenge in the treatment of patients with advanced PCa. For achieving significant progress in this area, one must explore the driving molecular mechanisms that underline PCa treatment resistance in order to identify new actionable targets.

### 1.2. Key Pathways Involved in PCa Growth and Disease Progression

PCa development and progression is the result of a complex multifactorial carcinogenesis process that involves several genetic and environmental factors. Among the deregulated molecular events occurring in PCa, AR signaling is historically considered the main carcinogenesis driver, playing a pivotal role both in tumor growth and development of castration-resistant disease [21]. In normal conditions, AR mediates the transcription of androgen-responsive genes, including cell proliferation and survival regulators, thus promoting the physiological activity and growth of the prostate epithelium. During PCa progression, cell growth and survival strictly depend on androgens [21]. In this context, the maintenance of AR signaling has been attributed to several dysfunctional molecular mechanisms, including AR gene amplification, mutations in AR gene sequence, and also changes in the expression of AR co-factor regulators. Alterations of AR signaling are rare in primary PCa but frequently found in CRPC, suggesting that such alterations are dependent of a selective pressure induced by ADT [22]. Overall, both AR amplification and mutations lead to the overexpression of AR protein, resulting in AR signaling activation even in presence of low circulating levels of androgens due to ADT. Interestingly, AR mutations also include AR mRNA splice variants (AR-Vs), which cause the production of trunked forms of AR. Some of them—e.g., AR-V7—are constitutively active independently from androgens, conferring to PCa a more aggressive phenotype by fostering castration-resistant cell growth [22,23]. Clinically, the detection of AR-Vs has been associated with resistance to abiraterone and enzalutamide in PCa patients [24].

Additional deregulated molecular mechanisms—other than AR signaling—have been associated to PCa growth and disease progression, including DNA repair pathways, PTEN/PI3K/AKT/mTOR signaling, and WNT pathway.

Defects and mutations in key genes of DNA repair pathways, including important regulators of the double-strand DNA break machinery, such as the homologous recombination repair (HRR) genes BRCA1 and BRCA2 and the DNA damage checkpoint activator ATM, have recently been reported in PCa. Interestingly, in carriers of BRCA1, BRCA2, or mismatch repair (MMR) germline mutations, an increased risk of developing PCa has been observed compared to whole population [22,25,26,27]. From a clinical point of view, the identification of defects in DNA repair pathways has provided a robust rationale for developing a therapeutic strategy on the basis of the use of DNA damaging agents in combination with PARP inhibitors. In this regard, when DNA repair gens are compromised, the pharmacological inhibition of PARP1 and PAPR2 leads to cell death according to the synthetic lethal model [28].

PTEN/PI3K/AKT/mTOR signaling are the most commonly altered pathways in primary PCa and also in about the 50% of CRPC [22]. Inactivation of PTEN, by bi-allelic deletion or hotspot mutations, frequently occurs in PCa and results in a constituting activation of PI3K/AKT/mTOR signaling, which plays a key role in the regulation of apoptosis, cell cycle progression, cellular proliferation, metabolism, differentiation, and migration [29]. Clinically, loss of PTEN functions has been recognized as a critical event in the progression from hormones native to CRPC.

Aberrations driving PCa development also involve the WNT pathways, which are generally inactivated in normal conditions. Mainly due to hotspot-activating mutations of β-catenin-encoding gene and APC promoter hypermethylation, WNT signaling is frequently altered in mCRPC [22]. This pathway plays a role in the regulation of several cellular events that are strictly related to metastasis development, including cell adhesion, migration, and epithelial-to-mesenchymal transition activation [30].

### 1.3. Molecular Contributors Underlying PCa Treatment Response

Advances in understanding the heterogeneity and molecular complexity of PCa could pave the way for the identification of novel strategies able to delay, or mitigate, the occurrence of adverse outcome in PCa patients.

#### 1.3.1. Radiotherapy

Among the putative mechanisms involved in PCa response to RT, DNA damage repair (DDR) system and oxidative stress are certainly the main actors linked to radiation response [31].

It has been largely reported that deregulated DDR mechanisms and elevated intracellular levels of reactive oxygen species (ROS) can significantly contribute to boost proliferation and self-renewal of PCa cells, thus allowing radiation stress to be overcome. The primary and direct effect of radiotherapy is the induction of DNA breaks, including both single-strand and double-strand breaks, resulting in cell cycle arrest and apoptosis. Alterations, mutations, and aberrant expression of genes strictly involved in the DDR system and checkpoints (i.e., p53/MDM2, ATM/ATR) and programmed cell death signaling (i.e., PTEN/Akt, Bax/Bcl-2, PARP-1) have been frequently reported in PCa and associated with adverse outcome to radiotherapy [32,33,34].

Additionally, a crucial indirect action of radiation is related to ROS production, being highly reactive and able to damage several molecules, including DNA and mitochondria, thus ultimately inducing local and systemic oxidative stress and consequent cell [35,36]. Given the important role of oxygen in ROS generation upon radiation, the status of tissue hypoxia plays a determinant role in treatment response. In fact, PCa hypoxic tumors characterized by a deregulation of hypoxia-related factors, such as HIF, PI3K/Akt/mTOR, NOX, Wnt/β-catenin, and Hedgehog, have been associated with adverse outcome after treatment with radiotherapy [37].

An additional process extensively involved in radiation response is epithelial-to-mesenchymal transition (EMT), which has been reported to confer radiotherapy resistance in PCa [38,39]. Several EMT-inducing molecules, including TGF-β, Wnt, Hedgehog, Notch, EGFR/PI3K/Akt, MAPK, and p21-PAK1, were found to be upregulated in the radioresistant phenotype [38]. In addition, mesenchymal proteins, such as Snail, Vimentin, and ZEB-1, were identified as putative biomarkers in PCa patients who develop radiotherapy resistance [40,41]. It is worth noting that among the aforementioned genes, an interesting molecule at the crossroads between different cellular pathways involved in radiotherapy response is represented by PARP-1. Specifically, it has been reported that PARP-1-overexpressing PCa cells can overcome radiation stress by undergoing phenotypic reprogramming through EMT. This piece of evidence highlights the strict and intricate interconnection existing between the various pathways, contributing to determining cell response to radiation [42].

Even if these mechanisms have been partially dissected and extensively studied through the use of sophisticated experimental models, PCa radiation response is still highly an unpredictable event [31].

#### 1.3.2. Androgen Deprivation Therapy

Response to ADT largely depends on the AR status. Upon ADT, tumor cells undergo a dramatic apoptosis [43]. Eventually, a subpopulation of cells can compensate reduced androgen levels by overexpressing AR in the presence of low androgen levels [44]. This mechanism represents the main adaptive strategy against ADT in PCa cells. However, alternative pathways can stimulate and promote AR activity. For instance, the NF-κB transcription factor can activate cellular pathways able to induce AR protein overexpression, overcoming the dependence of AR activation from androgens [45]. In addition, PTEN loss, which is one of the most frequent alterations found in PCa, can promote tumor growth independently from AR signaling [46]. Finally, it has been proved that the growth factors IGF-1 and EGR are also able to directly active AR independently from androgen systemic levels [47]. However, it is now clear that AR variants can also play a central role in ADT response and are widely involved in the AR ligand-independence activity [48]. Among the constitutively active AR variants, the AR-V7 has been identified as a driver of resistance acquisition to the second-generation AR-directed therapies. Interestingly, clinical evidence suggests that AR-V7 variant is involved in the induction of abiraterone acetate and enzalutamide resistance, but not in chemotherapy regimen based on the use of taxanes [24,49].

#### 1.3.3. Chemotherapy

Since the relatively poor approval period in PCa treatment, some mechanisms of response to taxane-based chemotherapy have already been identified. Taxanes act by stabilizing microtubules, resulting in apoptosis induction and G2/M arrest in cells. In PCa cells, taxanes can also interfere with AR trafficking within the nucleus, which is dependent on microtubule polymerization [50,51,52]. One of the main studied molecular mechanisms involved in the acquisition of resistance to taxanes is related to the overexpression of ATP-binding cassette (ABC) transporters in tumor cells [53], which induces a substantial drug efflux from the cells. Notably, due to the weak affinity for ABC transporter, cabazitaxel seems to be less susceptible in developing resistance compared to docetaxel [54]. In addition, the increase of the structural isoform of β-tubulin (III-β- tubulin) has been reported to reduce the affinity of docetaxel to microtubules [55]. Finally, taxane response is further associated with altered expression of EMT markers. Decreased levels of E-cadherin and concomitant upregulation of mesenchymal markers in tumor cells are involved in the regulation of cabazitaxel activity, highlighting that resistance to the taxane occurs in more mesenchymal and invasive cell phenotype. Interestingly the blocking of EMT process may substantially modulate taxane-based chemotherapy response [56].

### 1.4. MicroRNAs in Cancer

microRNAs (miRNAs) are short (18–25 nucleotides) non-coding RNAs. Thus far, almost 50 thousand mature miRNAs have been discovered, according to the latest release of the miRBase catalog [57] (http://miRbase.org (accessed on 13 January 2021)). miRNAs epigenetically regulate gene expression at the post-transcriptional level, warranting prompt and reversible alterations in protein expression though not affecting transcription by specifically recognizing and binding complementary sequences within target mRNAs. Depending on the degree of target complementarity, miRNAs accomplish negative gene regulation by two alternative mechanisms. miRNAs engaged in perfect complementarity binding induce target cleavage and direct degradation. For most mammals, miRNAs bind their targets through partial complementarity, inducing translation repression and consequent decrease in target gene protein levels [58,59].

Given that a perfect base pairing is required only between the target and the first 2–8 bases at the 5′ end of the miRNA (seed sequence) to guarantee efficient silencing, an individual miRNA can potentially target and regulate several transcripts and, in turn, each transcript can be targeted by different miRNAs [60]. This intrinsic peculiarity of miRNAs allows them to coordinate the expression of hundreds of genes, putting this class of molecules at the crossroads of a wide range of biological processes, including cell development, differentiation, metabolism, proliferation, cell cycle, and apoptosis [60]. Considering their involvement in all these crucial cell phenomena, the deregulation of miRNAs is causatively associated with the onset of several pathogenic conditions, including cancer. In this regard, miRNAs were reported to take part in pathways belonging to all the hallmarks of cancer by promoting or impairing tumorigenesis depending on their expression levels in specific cellular contexts and on the functions of their main targets [60]. For instance, miRNAs that negatively modulate the expression of oncogenes exert an oncosuppressive function and are generally downregulated in cancer. Conversely, miRNAs inhibiting oncosuppressor genes act as oncogenic molecules (oncomiRs) and are hence instrumentally over-expressed in tumor cells to ultimately sustain tumorigenesis [61].

miRNAs are also part of a regulatory circuit with other epigenetic factors. Specifically, the expression of miRNAs can be regulated by chromatin condensation status, depending on DNA and histone methylation as well as histone acetylation. In the other hand, these mechanisms are under the control of miRNAs in a feedback regulatory loop. Epigenetic modifications and aberrant miRNA expression can lead to the disruption of this feedback network facilitating the development and progression of cancer [62].

On the basis of the ambivalent role of miRNAs as anti- or pro-tumor factors, researchers can exploit the regulation of miRNAs to influence their target expression and consequently modulate downstream signaling pathways, with the ultimate aim of intercepting tumor progression. To this purpose, miRNA expression can be artificially modulated through opposite approaches. Specifically, the suppression of upregulated oncogenic miRNAs is achieved through the use of “antisense” oligonucleotides able to bind and dampen specific miRNAs. Conversely, miRNA mimics, small exogenous RNA molecules that mimic endogenous miRNAs, are largely exploited to restore the expression of downregulated oncosuppressor miRNAs [63].

### 1.5. MicroRNAs in Prostate Cancer

On the basis of the possibility to ectopically interfere with tumor-associated pathways that are under miRNA regulation, researchers’ interest in this large class of molecules, as novel targets or tools in miRNA-based anti-tumor strategies, exponentially increased in the last decades. Interestingly, miRNAs have been largely implicated in the regulation of several processes underlying the development of different types of cancer, including PCa [64]. Dozens of miRNAs have been reported as deregulated in PCa tissues and cells compared to normal prostate, and for many of them a functional role has been described [65].

Most frequently, PCa-related miRNAs are reported to behave as tumor suppressors, which is in line with the evidence that most deregulated miRNAs are downregulated in the disease compared to normal prostate tissue [66]. The tumor-suppressive role of miRNAs has generally been ascribed to their downregulation in PCa samples compared to normal counterparts, along with their ability, upon ectopic reconstitution, to interfere with cell proliferation, migration, and invasion, as well as to promote cell apoptosis, through the downregulation of specific targets involved in the aforementioned processes [61].

The first miRNAs reported to exert oncosuppressive functions in PCa are miR-15 and miR-16 [67]. In 2008, Bonci and collaborators demonstrated that the miR-15a–miR-16-1 cluster affects prostate cell survival, proliferation, and invasion by regulating cyclin D1 and WNT3A expression. Ectopic expression of miR-15a and miR-16-1 in PCa xenografts was found to induce significant tumor growth inhibition, and, consistently, the suppression of the two miRNAs through specific antisense oligonucleotides fostered the growth potential of normal prostate cells both in vitro and in vivo [67] (Table 1).

In terms of relevance for the disease, the first-in-class miRNA in PCa is undoubtedly miR-205, which was extensively reported to be downregulated in PCa tissues and cell lines compared to adjacent non-neoplastic tissue and normal prostate cells, respectively [70]. In 2009, Gandellini et al. showed that restoration of miR-205 in aggressive PCa cells induced marked morphological changes and cytoskeleton rearrangements, which were compatible with a reversion of EMT [68]. This evidence was supported by the increase in expression and membrane localization of E-cadherin together with the repression of several factors involved in the acquisition of an invasive behavior (i.e., IL-6, EZH2, caveolin-1, and MMP-2). The observed involvement of miR-205 in maintaining the epithelial organization of human prostatic tissue by impairing the EMT process was demonstrated to be mainly driven by the concurrent suppression of relevant miR-205 targets, including PKCε [68,69]. The crucial role of miR-205 in PCa EMT was further corroborated by evidence that this miRNA was the most downregulated miRNA in PCa cells when stimulated by cancer-associated-fibroblasts (CAF). Rescue experiments showed that ectopic miR-205 overexpression in PCa cells counteracts CAF-induced EMT, thus impairing enhancement of cell invasion, acquisition of stem cell traits, tumorigenicity, and metastatic dissemination (Table 1). In addition, miR-205 was shown to interfere with tumor-driven activation of surrounding fibroblasts by reducing pro-inflammatory cytokine secretion [77]. Additional studies showed that, in the normal human prostate, miR-205 is physiologically involved in the regulation of basement membrane deposition by participating in a circuit involving ΔNp63α, which is known to be essential for the maintenance of the basement membrane in prostate epithelium. At the molecular level, the researchers showed that ΔNp63α promotes miR-205 transcription by binding to its promoter, and the miRNA, in turn, reduces ΔNp63α protein expression, acting at the level of proteasomal degradation, ultimately leading to the restoration of the basement membrane deposition [78]. These findings provided a further mechanism by which miR-205 is potentially implicated in hampering PCa tumorigenesis.

In these regards, Profumo et al. demonstrated that not only miR-205 but also its host gene MIR205HG has a crucial role in preserving the basal identity of prostate epithelial cells by repressing the transcription of its targets and likely buffering the activity of IRF1. By inhibiting IRF-1, MIR205HG substantially contributes to preventing the luminal differentiation of the prostate epithelial cells, acting independently from miR-205, albeit complementing its role in maintaining prostate epithelial phenotype [79].

Along with miR-205, among the miRNAs consistently reported as downregulated in PCa is miR-34a. miR-34a is one of the most widely studied oncosuppressive miRNAs in cancer [80] and the first one to be exploited for a micro-RNA-based clinical therapy. A first-in-human phase 1 study of MRX34, a liposomal miR-34a mimic, has been carried out in patients with advanced solid tumors, some of whom experienced partial response and long-lasting stable disease. The study was terminated due to serious immune-mediated adverse effects [81]. In keeping with its primacy as oncosuppressive miRNA, miR-34a was found to be significantly downregulated in many human tumors, including PCa, where mainly pro-apoptotic functions have been ascribed to the miRNA. Indeed, different studies have reported that ectopic miR-34a expression induces apoptosis, cell cycle arrest, and senescence, as well as inhibiting cell growth in PCa cells through repression of different targets, including SIRT1 [71], Wnt1 [72], c-Myc [73], and STMN1 [74] (Table 1). Liu et al. showed that miR-34a was downregulated in PCa CD44+ cells, which are characterized by enhanced clonogenic, tumor-initiating, and metastatic capabilities. Enforced expression of miR-34a in these cells reverted their characteristics by inhibiting clonogenic expansion, tumor regeneration, and metastasis, while expression of miR-34a downregulation by antisense oligonucleotides promoted tumor development and metastasis. Moreover, knockdown of CD44, validated as a direct target of miR-34a, was shown to inhibit PCa growth and metastasis, thus phenocopying miRNA effects [82]. More recently, Dong and collaborators reported that miR-34a is significantly downregulated in PCa compared to paired normal tissues. Moreover, miRNA reconstitution in PC3 cells negatively affected the Wnt/β-catenin pathway by directly targeting Wnt1, hence inhibiting cell proliferation and migration and promoting apoptosis. This evidence suggests that miR-34a acts as an oncosuppressive miRNA also in the context of PCa, thus further encouraging the application of a miR-34a-based therapy in a wide range of cancers, including PCa [66].

Unlike miR-34a, the story starring miR-21 talks about a miRNA for which controversial results have been reported in the context of PCa, despite its undisputed role as an oncomiR in several tumor types. Indeed, miR-21 is a widely studied miRNA found to be overexpressed in numerous human cancers, where it has been reported to be endowed with oncogenic properties due to its ability to negatively modulate the expression of tumor suppressor genes and hence to cause the reversion of malignant phenotype when knocked down in several tumor models [83,84,85]. On the basis of this evidence, researchers have referred to miR-21 as an oncomiR potentially exploitable as promising target for anticancer therapy. However, in the context of PCa, miR-21 has shown opposing results. In line with its reported oncogenic role, Guan et al. found that miR-21 was highly expressed in PCa cells and its overexpression promoted PCa cell proliferation, migration, invasion, and resistance to apoptosis by directly targeting KLF5 and consequent upregulation of GSK3B and activation of the AKT signaling pathway [75] (Table 1). Consistently, Yang et al. reported that miR-21 overexpression in PC3 cells inhibits PTEN, thus promoting PCa cell proliferation and invasion [76] (Table 1). However, an interesting analysis has been advanced by Folini et al. who stumbled across different results while investigating the effects of miR-21 in PCa [86]. Upon miR-21 downregulation in cell lines highly expressing the miRNA and characterized by a different status of PTEN, these authors showed that miR-21 knockdown is not sufficient per se to significantly modify the proliferative potential of PCa cells. Specifically, they found that, even in PTEN positive DU145 cells, the downregulation of miR-21, achieved through specific antisense oligonucleotides, was not able to substantially alter PTEN expression levels, suggesting that miR-21 is not involved in the regulation of PTEN expression in DU145 cells. Moreover, miR-21 was not found to be differently expressed in prostate carcinomas compared to matched normal tissues, further suggesting that miR-21 does not perform as an oncomiR in PCa [86]. In interpreting and discussing their findings, the authors argue that the oncogenic properties imputable to miR-21 could be extensively influenced by the relative and mutual expression of the miRNA and of its targets, both factors implicated in determining the ultimate function of a specific miRNA in a specific cell context. For instance, a concomitant unbalanced deregulation of different miRNAs with opposed function, converging on the same target genes thus producing divergent effects, could lead cancer cells to compensate the antitumor phenotype induced by the miRNA in different experimental models. Conversely, the simultaneous downregulation of oncogenic miRNAs regulating the same biological processes through the downregulation of different targets could synergize in counteracting the proliferative potential of PCa cells. The proposed argument, besides articulating the possible mechanisms underlying the dual nature of miR-21 in different tumor settings, covers a central issue regarding miRNA studies, underlying the relevance of properly contextualizing the properties ascribed to specific miRNAs by taking into account their intrinsic ability to regulate multiple targets in a cell and tissue dependent fashion.

## 2. miRNAs Involved in PCa Response to Ionizing Radiation

It is well established that ionizing radiation (IR) damages cancer cells by producing intermediate ions and free radicals, causing different types of damage including DNA single-strand breaks (SSBs) and double-strand breaks (DSBs), the latter being the most lethal form of DNA damage. Both SSBs and DSBs trigger the DNA damage response (DDR), an intricate and sophisticated network of signaling pathways aimed at fixing the induced damage to overcome the injury, or toggling to cell death, whereby the damage is unrecoverable. Interfering with DDR or downstream signaling pathways can significantly improve the impact of IR on cells, ultimately enhancing RT effectiveness [87].

The intrinsic cell sensitivity to radiation relies on several factors acting in cellular pathways relevant to radiation response. By regulating the expression of these factors, miRNAs play a main role in the radiation response of many tumors, including PCa [64]. One approach to explore the role of miRNAs in radiation response is studying their modulation upon cell exposure to IR [88]. In this regard, studies comparing normal and cancer cell models using genome-wide analysis through microarray-based screenings or the evaluation of wide miRNA panels by qRT-PCR have shown that IR is able to significantly alter the expression levels of several miRNAs. However, only a partial overlap between reported modulated miRNAs has emerged from the literature, showing different results across cell lines and also within the same cell line throughout various conditions, suggesting that miRNA profiles change upon radiation exposure following a spatio-temporal dynamic governed by cell and experimental condition specificity [89,90]. Notably, the evidence that miRNAs are largely modulated upon radiation is not sufficient per se to warrant a causal role of miRNAs in radiation response. To ascribe a functional involvement of specific miRNAs in PCa radiation response, their ability to actively influence cell sensitivity to radiation should be proved by gain- or loss-of-function studies where miRNA expression is ectopically reconstituted or inhibited in cell models through specific miRNA mimics or inhibitors, respectively. Upon artificial modulation, a functional role in radiation response is assigned to the miRNA when its capability to positively or negatively influence cell sensitivity profile to radiation, through interference with specific mechanisms that are known to be relevant to radiation response, is observed [91]. Indeed, miRNAs are involved in many biologic mechanisms that can impact cell response to radiation by either promoting or impairing cell proficiency to recover from the induced damage. The different biological events in which miRNA involvement has been consistently reported to affect PCa radiation response ultimately converge on key cell processes including DDR, cell survival/proliferation, apoptosis/autophagy, cell cycle checkpoint, and EMT [64,89]. miRNAs reported to directly affect radiation sensitivity of PCa models by regulating factors involved in these mechanisms are discusses herein and are reported in Figure 1 and Table 2.

### 2.1. DNA Damage Response

DDR is a vital process triggered by IR, responsible for maintaining genomic integrity by sensing and recovering from the DNA damage induced by radiation through the activation of diverse DNA repair systems on the basis of the nature and the entity of the damage. In this scenario, different miRNAs have been reported to interfere with DDR efficiency by regulating genes directly involved in the DDR system, thus ultimately inducing either radiation sensitivity or radiation resistance.

In a comprehensive study aimed at identifying DNA damage-regulating miRNAs by screening a library of 810 miRNA mimics for the ability to influence cellular sensitivity to IR, the researchers found a great number of miRNAs to increase cell sensitivity to IR in a PCa luciferase cell model [108]. Two miRNAs identified as the most potent sensitizers, miR-890 and miR-744–3p, were shown to significantly delay radiation-induced damage repair by inhibiting the expression of multiple components of DDR. Specifically, miR-890 directly targeted MAD2L2, WEE1, and XPC, whereas miR-744–3p directly targeted RAD23B. Interestingly, the knockdown of individual miR-890 targets by phenocopy experiments was not sufficient to abrogate miR-890 radiosensitization, suggesting that miR-890 exerts its radiosensitizing effect by regulating multiple DDR genes. Moreover, miR-890 reconstitution was able to enhance the therapeutic effects of radiation delivered to animal models, confirming miR-890 as a potential IR sensitizing agent [108].

Other miRNAs modulated by IR showing a causal role in radiation response by affecting DDR are miR-99a and miR-100, belonging to the miR-99 family, which were reported to be downregulated in radioresistant PCa and upregulated following IR exposure. The ectopic overexpression of both miRNAs in PCa cells reduced the efficiency of DSB repair and increased cell radiosensitivity by targeting SNF2H. Inhibition of SNF2H hindered the recruitment of BRCA1 at the DNA damage sites, thus preventing the initiation of homologous recombination (HR) pathway and thereby augmenting cell sensitivity to radiation [111].

Recently, miR-205 has been reported to considerably improve PCa radiation response by hindering radiation-induced DDR through the inhibition of PKCε and EGFR nuclear translocation, as well as causing consequent activation of DNA-PK, a major determinant in the non-homologous end-joining (NHEJ) DDR pathway [69].

Another miRNA that has been steadily reported to exert a radiosensitizing effect in PCa by impairing DDR is miR-145. Gong and collaborators demonstrated that miR-145 reconstitution sensitizes PCa cells to radiation and impairs the efficiency of radiation-induced DSB repair, as indicated by the increased foci of γ-H2AX, a surrogate marker of DSBs, and reduced expression of 10 DDR-related genes. Consistently, miR-145 was found to be overexpressed in radiotherapy-responsive patients, while miR-145-regulated DDR genes were significantly downregulated [96]. In support of its radiosensitizing potential in PCa, more recently, El Bezawy et al. demonstrated that miR-145 ectopic expression, resulting in direct inhibition of SPOP, the most commonly mutated gene in PCa, was able to enhance radiation response of PCa both in vitro and in vivo by downregulating RAD51 and CHK1, key players in the HR DDR pathway [97].

The enhancement of radiation sensitivity of PCa cells upon DDR regulation was described also for miR-521, a miRNA whose expression was found to be significantly downregulated in PCa cells exposed to radiation. To evaluate a potential functional role of miR-521 in PCa radiation response, Josson et al. overexpressed the miRNA through the use of a specific mimic and observed that miR-521 overexpression sensitized PCa cells to radiation treatment. Consistently, ectopic inhibition of the miRNA resulted in radiation resistance of PCa cells. The effects of miR-521 on radiation response were ascribed to the modulation of its target CSA, a DNA repair protein [106] (Figure 1 and Table 2).

### 2.2. Cell Cycle Checkpoints

The efficiency of the DDR system in repairing IR-induced DNA damage determines whether the cell is ready or not to proceed along the cell cycle. In order for the decision to be made, cell progression is arrested at G1 and G2 checkpoints to allow time for checking the status of DNA damage. Typically, only once DNA repair is ascertained can the cell re-enter the cell cycle [89]. The efficient transition throughout the cycle is guaranteed by different checkpoints that serve as potential termination points, ensuring that progression to the next phase of the cell cycle occurs only when favorable conditions are met. Progression through these checkpoints largely depends on cyclins, cyclin-dependent kinases (CDKs), CDK inhibitors, and transcription factors, and each of these components can be subjected to miRNA regulation.

In PCa irradiated cells, miR-16-5p was reported to induce cell cycle arrest at G0/G1 phase by targeting cyclin D [98]. Specifically, the authors found that miR-16-5p was significantly upregulated in PCa LNCaP cells upon IR delivered by either X-rays or heavy ions, and its overexpression inhibited cell proliferation and induced cell cycle arrest at G0/G1 phase by directly targeting the cyclin D1/E1-3′-UTR, thus enhancing LNCaP cell radiosensitivity. An opposite effect on radiation response following cell cycle arrest was observed upon miR-106b overexpression in LNCaP cells [92]. In this regard, Li et al. showed that miR-106b was the most downregulated among several miRNAs deregulated upon exposure to IR. Ectopic expression of miR-106b in LNCaP cells suppressed the radiation-induced increase of p21, suggesting that miR-106b is able to override radiation-induced cell cycle arrest in G2/M and promote cell proliferation. Another miRNA reported to overcome radiation-induced arrest in G2/M phase is miR-95. This miRNA was found to be significantly upregulated in irradiated PCa PC-3 cells compared to untreated controls, and its enforced expression promoted cell transit through G2/M phase by directly inhibiting SGPP1, consequently leading to increased cellular proliferation and promoting radiation resistance both in vitro and in vivo [110].

Contrasting results have been reported for miR-449, a miRNA that, by inducing cell cycle arrest at G2/M phase, was demonstrated to enhance PCa radiosensitivity in both cell and animal models [102]. Specifically, Mao and collaborators found that ectopic expression of miR-449, a miRNA previously reported to be downregulated in prostate cancer tissues compared to patient-matched control tissues [103], enhanced radiation-induced G2/M phase arrest by modulating pRb/E2F1, and this resulted in augmented cell apoptosis and consequent sensitization of prostate cancer cells to IR [102]. Later on, they studied the effect of miR-449 on the radiation response in LNCaP cell model, where they found that the miRNA was upregulated upon IR, and its ectopic overexpression, in turn, was able to increase radiosensitivity in LNCaP cells and xenografts by promoting radiation-induced G2/M phase arrest consequent to the direct suppression of c-Myc, a novel target of miR-449 that they found to also be consistently repressed in miR-449-transfected PC-3 and DU145 cells [104].

miR-107, a member of the miR-15/107 “superfamily” [112], has been recently appended to the list of miRNAs functionally involved in the radiation response of PCa. In this regard, Lo et al. found that miR-107 is downregulated in response to radiation in PC-3, and its ectopic overexpression is able to significantly increase cell radiosensitivity by directly targeting the pleiotropic growth factor GRN, as supported by phenocopy (by GRN suppression) and rescue (by GRN overexpression) experiments that showed a recapitulation or an attenuation of miRNA-induced radiosensitization, respectively. Mechanistically, miR-107 was shown to alter cell cycle distribution, promoting radiation-induced G1/S phase arrest and G2/M phase transit, as well as enhancing delayed apoptosis through suppression of p21 and CHK2-phosphorylation [93] (Figure 1 and Table 2).

### 2.3. Apoptosis and Autophagy

miRNAs can increase radiosensitivity by affecting several cell processes that at the end likely converge on the inhibition of cell proliferative potential and induction of cell death via multiple mechanisms [113,114]. The mode of cell death following IR is determined by a wide variety of factors such as cell type and radiation modality. In PCa, apoptosis seems not be the predominant mechanism for radiation-induced cell death; rather, the permanent growth arrest, resembling a senescent-like phenotype, was proposed as the major responsible of radiation-induced clonogenic cell death [115]. Whatever the underlying mechanism, miRNAs may directly or indirectly lead to cell death/growth arrest, thus ultimately altering cancer radiosensitivity.

In the context of miRNAs influencing PCa radiation response by regulating apoptosis, miR-541-3p has been described very recently [107]. miR-541-3p, a miRNA that is poorly expressed in PCa tissues compared to normal controls, is overexpressed upon radiation in PCa cells. In a loss-of-function setting, miR-541-3p knockdown increased the proliferative potential and decreased the apoptotic rate of irradiated cells, ultimately reducing cell radiosensitivity. Conversely, miR-541-3p overexpression by miRNA mimic increased cell sensitivity as a result of a reduction in cell viability and colony formation, paralleled by increased apoptosis. Mechanistically, HSP27, validated as a direct target of the miRNA, was proposed as the potential mediator of miR-541-3p-induced radiosensitization, as suggested by rescue experiments showing a partial reversion of miRNA biological effects upon HSP27 ectopic overexpression. Another miRNA affecting IR-induced apoptosis is miR-498 [105]. When overexpressed by mimic transfection, miR-498 was shown to improve LNCaP and DU145 cell proliferation and survival, inducing radioresistance. Conversely, miRNA suppression through specific inhibitors negatively affected cell proliferation and survival, thus improving cell response to radiation. Target-identification analysis revealed that the expression of PTEN, a validated direct target of miR-498, is inversely correlated with that of the miRNA, suggesting a potential involvement of this well-known tumor suppressor as a possible mediator of miR-498-induced radioresistance as a consequence of apoptosis impairment. Interestingly, miR-498 was also found to regulate several EMT-related proteins, including E-cadherin, N-cadherin, snail, and Vimentin, suggesting that a cross-regulation of multiple biological functions could be envisaged at the basis of miR-498 role in cell response to radiation.

Autophagy, a catabolic process guaranteeing the turnover and clearance of damaged proteins and organelles, is an important cytoprotective process that occurs in response to damaging stimuli, including irradiation [116]. Consistent with its contribution in sustaining cell survival, autophagy has been largely reported to prompt radioresistance bypassing radiation-induced cell stress. In this context, miR-32, found to be upregulated in PCa tissues and cell lines compared to normal counterpart, was shown to enhance tumor cell survival and decrease radiosensitivity in the PCa cells by inducing autophagy as a result of direct suppression of the autophagy-related protein DAB2IP [101]. Conversely, miRNA inhibition reverted these effects, resulting in increased radiosensitivity, rescuing DAB2IP expression levels and suppression of IR-mediated autophagy. Functional studies in PCa models showed that miR-124 and miR-144 inhibit autophagy and enhance radiosensitivity by simultaneously downregulating PIM1 kinase [94].

Similarly, miR-205 has been shown to concur with miR-30a to the inhibition of the shared target TP53INP1, an autophagy-related protein whose overexpression has been previously reported to correlate with PCa poor prognosis and to predict biochemical relapse [117], thus inducing a radiosensitizing effect via autophagy inhibition [100]. These findings have been recently confirmed for miR-205 by Wang and collaborators in a study revealing that the miRNA suppresses autophagy and enhances radiosensitivity of PCa cells by targeting TP53INP1 [99] (Figure 1 and Table 2).

### 2.4. Epithelial-to-Mesenchymal Transition

Epithelial-to-mesenchymal transition is a phenotypic switch that promotes the loss of epithelial characteristics, such as basal–apical polarity, cell–cell adhesion, and basal–lamina attachment, as well as the acquisition of a fibroblast-like morphology, defined by increased migratory and invasive properties, metastatic potential, and resistance to conventional treatments, including radiotherapy [118]. Given EMT contribution in determining cell response to radiation, miRNAs can actively participate in this process by targeting EMT-related genes.

In the scenario of miRNAs implicated in the regulation of EMT in the context of PCa, miR-205 performs as a main actor. As discussed above, miR-205 plays a pivotal role in regulating PCa EMT due to its ability to suppress EMT-related factors including PKCε [68,69]. Interestingly, El Bezawy et al. recently reported that miR-205 is able to enhance radiation sensitivity of PCa cells and xenografts through inhibition of both PKCε and ZEB1 [69]. ZEB1 silencing in phenocopy experiments was capable of reproducing miR-205 radiosensitizing effect, confirming that ZEB1 is functionally involved in miRNA-mediated radiosenzitizing effect. Concerning PKCε, its implication in miR-205-induced radiosensitization has been ascribed to its engagement in the nuclear translocation of EGFR. Indeed, PKCε induces EGFR accumulation in the nucleus, where it interacts with DNA-PK and increases its enzymatic activity through S2056 and T2609 phosphorylation, which are required for DNA DSB repair by the NHEJ pathway [69]. In keeping with this function, PKCε silencing, through specific a siRNA, was able to recapitulate the radiosensitizing effect induced by miR-205 and to reduce the accumulation of the phosphorylated forms of both EGFR and DNA-PK. Consistently, a target protection approach able to impair miR-205-PKCε physical interaction almost completely abrogated miRNA-radiosensitizing effect as a consequence of the complete rescue of PKCε expression levels [69].

El Bezawy et al. have also identified miR-875-5p as a novel miRNA able to enhance PCa radiation response by counteracting EMT [109]. Reconstitution of miR-875-5p, whose expression was downregulated in PCa clinical samples, was able to boost PCa radiation response through EGFR suppression. Consistent with its previously described implication in the DNA-PK–NHEJ axis, together with the evidence of its role in sustaining EMT [119,120], EGFR, here identified and validated as a direct target of miR-875-5p, was found to be significantly downregulated, and its nuclear translocation elicited by radiation was compromised upon miRNA reconstitution, thus hindering the activation of DNA-PK and hence dampening DNA lesion clearance. In the same study, miR-875-5p was demonstrated to downregulate ZEB1, a well-known EMT-inducing transcription factor, also reported to play a role in HR-mediated DDR by regulating CHK1. Interestingly, ZEB1 was found to be a major mediator of miR-875-5p-induced radiosensitization, as supported by phenocopy experiments showing that ZEB1 ectopic suppression is able to recapitulate the miRNA-induced enhancement of PCa sensitivity to radiation [109].

The ability of miRNAs to affect PCa radiation response by altering EGFR expression was also reported for miR-1272 [95]. The reconstitution of miR-1272 levels, which are decreased in tumor specimens compared to normal tissues, was able to revert the mesenchymal phenotype and affect the migratory and invasive properties of PCa cells. Consistent with its EMT-counteracting function, miR-1272 increased IR radiation response in PCa cells by indirectly reducing EGFR protein expression levels as a result of HIP1 direct targeting.

The evidence that specific miRNAs such as miR-875-5p and miR-205 exert their radiosensitizing function by concurrently acting at the level of different processes further substantiates the notion that miRNA ability to target multiple genes/pathways is instrumental in accomplishing specific functions by intervening in alternative mechanisms jointly concurring to the generation of a common biological outcome (Figure 1 and Table 2).

## 3. MiRNAs Involved in Drug Response

In the last decade, miRNAs emerged as crucial molecular regulators of several cellular pathways involved in PCa response and development of resistance to hormone therapy and chemotherapy. As regards ADT, the resistant phenotype has been frequently associated with the reactivation of AR signaling or the overcoming of androgen dependence by hiring additional cell survival pathways independent from AR functional activation [121].

Chemotherapy is the therapeutic option offered to mCPRC patients who have failed other treatment approaches and is mainly based on the use of taxanes (docetaxel, cabazitaxel). Resistance to these drugs is a multifactorial phenomenon in which alterations of microtubule regulatory proteins and tubulin isotopes play a major role [122].

It is well accepted that miRNAs can take part and influence the acquisition of the resistant phenotype to specific treatments in PCa, although their role in mediating castration resistance and chemotherapy resistance is still to be clarified thus far. However, the progressive shift from in vitro experiments on cell lines to the use of PCa clinical samples and in vivo models could certainly generate more translationable information on the potential of specific miRNAs as therapeutic tools/targets and predictive biomarkers to be implemented in the clinics for improving patients’ management. miRNAs reported to affect drug response of PCa models are discussed herein and reported in Figure 2 and Tables 3 and 4.

### 3.1. AR Signaling

miRNAs are involved in PCa response to ADT, mainly by modulating AR signaling and subsequently activating molecular pathways engaged in cell cycle deregulation. Among the miRNAs implicated in AR pathway regulation, miRNA-185 was found to be decreased in PCa tumors compared to normal tissues [123]. In this setting, luciferase reporter assay and target protector experiments showed that miR-185 can directly bind the 3′UTR of AR mRNA in PCa cells, resulting in decreased AR expression [124]. In addition, miR-185 can also attenuate the functional activation of AR by downregulating the AR co-activator bromodomain containing 8 isoform 2 (BRD8 ISO2), ultimately inducing its mRNA repression [125] and overcoming androgen dependence of PCa cells. Overall, the ectopically expression of miR-185 can consistently affect proliferation, migration, invasion, and in vivo tumorigenicity of PCa cells [123]. An interesting link between androgen deregulation and miR-185 was reported by Li and collogues. Specifically, miR-185 reconstitution can downregulate the expression of the oncogenic transcription factor sterol regulatory element-binding protein-1 (SREBP-1), thus avoiding lipid accumulation and inducing AR transcriptional activity deregulation, ultimately resulting in cell proliferation arrest and reduction of invasion capability and tumor growth in PCa models [126]. These results suggest that interfering with abnormal lipogenesis and cholesterogenesis could provide a promising and innovative therapeutic approach for the treatment of PCa.

The controversial role of miR-221/222 cluster in the progression from androgen sensitive PCa to CRPC has been investigated in several preclinical studies. Initially, miR-221/222 downregulation has been observed in metastatic PCa and CRPC tumors compared to normal tissues, suggesting its role as a putative tumor suppressor involved in the progression form androgen dependency to castration resistance [127]. However, a growing body of evidence has shown that the role of miR-221/-222 on PCa cell growth is certainly cell type-specific and context-dependent. Specifically, it has been shown that in AR-independent PCa cells, such as PC3 and DU145, miR-221/-222 reconstitution inhibits cell proliferation [128]. In contrast, the ectopic expression of miR-221/222 in AR-sensitive LNCaP cells promoted cell proliferation [129]. These findings were confirmed in in vivo PCa models, where miR-221/222 reconstitution in LNCaP cells strongly boosted LNCaP tumor growth while reducing PC3 tumor growth rate in mice [130]. Finally, Bin Gui and colleagues recently reported that miR-221/-222 are AR-repressed genes and their expression and function are dependent on AR status in PCa cells [127].

Additional miRNAs that belong to the miR-30 family can directly affect AR expression by binding the 3′UTR, the coding region of AR and ARv7 transcripts. Experimentally, Kumar and colleagues [131] found that ectopical reconstitution of miR-30b-5p or miR-30c-5p was sufficient to reduce AR protein expression in LNCaP and VCaP cells. In addition, the inhibition of endogenous miR-30a-5p, miR-30b-3p, miR-30c-5p, and miR-30d-5p performed in an androgen-deprived environment made PCa cell proliferation independent from androgens. Moreover, miR-30d-5p expression levels were found to be decreased and inversely correlated with AR in CRPC tumors [131].

The identification of specific AR variants (i.e., ARv7, ARv4) strictly associated with AR-directed therapy failure in PCa has led to dissect the molecular mechanisms underlying AR variant-induced resistance in PCa. In this regard, two miRNAs, miR-34c and miR-449b, were found to be involved in AR splicing variant regulation, directly suppressing both ARv7a and ARv4 expression [132].

Among the miRNAs associated with AR variant regulation, miR-124 can represses the AR expression variants along with the AR cofactor EZH2 by targeting the 3′UTRs of ARv7 and ARv4 transcripts. Interestingly, PCa xenograft-bearing mice were sensitized to ADT by systemic administration of miR-124, ultimately resulting in a significant reduction of the tumor growth [133] (Figure 2 and Table 3).

### 3.2. Survival Pathways/Apoptosis Escape

In tumor development and progression, miRNA deregulation has been largely associated with dysfunctional survival pathways, thus frequently resulting in cell cycle arrest and apoptosis escape. In this respect, miR-143, a miRNA found to be downregulated in PCa at both early and CRPC stages [134,135], has been extensively reported to be involved in such cellular events. miR-143 leaking in PCa cells was reported to cause the downregulation of ERK5, thus fostering cell proliferation and apoptosis escape [136]. Notably, the ectopic overexpression of miR-143 in PCa cells resulted in increased sensitivity to docetaxel both in vitro and in vivo [137]. The mechanism by which miR-143 confers sensitivity to docetaxel in PCa cells has been defined by Xu et al., showing that the KRAS pathways are targeted by the miRNA and its deregulation modulates the response to taxane-based chemotherapy influencing cell proliferation and survival.

Growing evidence has experimentally proven the active involvement of miRNAs in the regulation of pro- and anti-apoptotic proteins, ultimately dissecting the molecular mechanisms underlaying the link between miRNA deregulation and therapeutic resistance in PCa. Among them, miR-223-3p was found to reduce docetaxel-induced apoptosis in PCa cells, by directly targeting the transcription factor FOXO3, which is involved in the regulation of cell homeostasis and apoptosis [138]. Additionally, miR-323 upregulation was reported to induce resistance to docetaxel in PCa cells by directly targeting the tumor suppressor p73 [139]. By targeting YAP and SEC23A, miR-375 attenuates docetaxel-induced apoptosis in in vitro and in vivo PCa models [140]. Again, miR-148a mimic transfection in paclitaxel-resistant PC3 cells confers sensitivity to the taxane by regulating MSK1 expression, a downstream effector regulated by MAPK signaling [141].

miR-21 has been frequently found to be overexpressed in many tumor tissues and is associated with the tumorigenesis process [83,85,142,143]. Although the role of miR-21 in PCa is still controversial, its involvement in AR signaling has been reported. Indeed, miR-21 transcription levels are directly regulated by activation of AR, which binds the miR-21 promoter [144]. Additional mechanisms by which miR-21 interferes with drug response in PCa are associated with its capability to regulate cell survival pathways. It has been initially reported that miR-21 overexpression is able to impart androgen-independent PCa cell growth, both in vitro as well in vivo, ultimately inducing resistance to docetaxel [144,145]. In this regard, Shi et al. experimentally defined the molecular mechanism by which miR-21 can confer docetaxel resistant phenotype to PCa xenografts. The authors demonstrated that miR-21 negatively regulates the proapoptotic and neoplastic transformation inhibitor PDCD4 factor, thus resulting in apoptosis deregulation in cells [146].

miR-34a, a tumor suppressor miRNA, found to be downregulated in PCa compared tonormal tissues [147], is also functionally involved in treatment response. Specifically, miR-34a-enforced overexpression in PCas enhance chemotherapy-mediated apoptosis in drug resistant PCa cells, mainly by targeting the antiapoptotic proteins SIRT1 and Bcl-2, as well as their downstream pathways. Specifically, docetaxel-resistant PC3 and 22Rv1 cells displayed enhanced sensitivity to paclitaxel upon miR-34a reconstitution [148]. The mechanisms by which miR-34a can induce sensitivity to chemotherapeutics seem to be streaky with regards to p53 status, and as for many other miRNAs, this scenario corroborates the established notion that each miRNA function is cell context-dependent and molecular network-related. In this regard, Roklin et al. observed that the overexpression of miR-34a in p53 wild-type LNCaP cells is not sufficient to increase doxorubicin-induced apoptosis. However, the authors showed that the simultaneous reconstitution of miR-34a and miR-34c enhanced p53-mediated apoptosis upon doxorubicin treatment in these cells [149].

A similar mechanism was observed for miR-205. Enforced expression of this miRNA in PCa cells increased sensitivity to cisplatin and doxorubicin through the downregulation of the antiapoptotic protein Bcl-2 and consequent apoptosis induction [128,150]. Consistently, Bhatnagar et al. found that downregulation of miR-205 confers resistance to chemotherapy-induced apoptosis in PCa cells by negatively regulating Bcl-w [151] (Figure 2 and Table 4).

### 3.3. Epithelial-to-Mesenchymal Transition Induction

Consolidate evidence indicates that EMT activation contributes to the occurrence of drug resistance in several cancers, including PCa. Among the miRNAs consistently associated with EMT regulation, miR-200 family members were widely studied as suppressor of EMT and cell spreading during PCa progression [152,153]. An interesting link between EMT activation and docetaxel-resistant PCa has been initially reported by Puhr et al., who showed that the expression of miR-200c—a miR-200 family member—is reduced in decetaxel-resistant PCa cells characterized by a more mesenchymal phenotype compared to parental cells [154]. Mechanistically, miR-200c ectopic expression was able to repress the EMT markers ZEB1 and ZEB2, and also enhance E-cadherin expression in docetaxel-resistant PCa cells, thus reverting EMT and ultimately resulting in an increased drug-induced apoptosis [154]. In addition, the capability to confer docetaxel sensitivity in PCa cells, as the result of EMT reversion, has been reported in relation to another miR-200 family member, miR-200b. In this regard, functional studies highlighted that miR-200b reconstitution increased sensitivity to docetaxel in PCa cells by fostering Bim-1 activation and finally inducing cell apoptosis [155].

Similarly, as previously discussed, miR-205 is one of the main EMT-related miRNA in PCa [68]. Puhr at al. reported that miR-205 reduced expression is associated with docetaxel resistance in PCa cells as a result of miRNA-mediated EMT impairment [154].

Finally, restoration of miR-128, which was found to be downregulated in PCa displaying an aggressive phenotype compared to normal tissues [156], was sufficient to sensitize PCa cells (DU145 and LNCaP) to cisplatin and to impair cell invasion capability [157]. Indeed, the authors demonstrated that miR-128-overexpressing PCa cells become sensitive to cisplatin as a consequence of the downregulated expression of the EMT master regulator ZEB1, which is directly targeted by the miRNA (Figure 2 and Table 4).

### 3.4. Drug Efflux Transporter Activity

Among the several deregulated mechanisms observed in cancer, the upregulation of ABC transporters has been described as a crucial event in chemotherapy response. By inducing an active extrusion of a board spectrum of drugs, ABC transporters can cause a substantial decline of chemotherapeutics activity in tumor cells. Recent evidence has highlighted the involvement of miRNAs and long non-coding RNAs (lncRNA) in such processes. In this context, it has been found that miR-204 and miR-34a are involved in a tangled deregulated circuit that influences PCa cell sensitivity to taxane-based drugs. Specifically, Jiang and collogues showed that the lncRNA NEAT1 induces docetaxel resistance in PCa cells by sponging miR-204 and miR-34a, thus resulting in the upregulation of ABCG2 and ABCC4 transporters via ACSL4 activation [158]. Moreover, miR-34a—a miRNA frequently downregulated in drug-resistant PCa—can be repressed by another lncRNA, DANCER [159]. By sponging miR-34a, DANCER contributes to releasing JAG1 and finally fostering docetaxel resistance by enhancing the expression of ABCB1 and ABCC4 transporters (Figure 2 and Table 4).

## 4. MiRNAs Involved in Neuroendocrine PCa Development

Recent evidence indicates that specific miRNAs are directly associated with the development of NEPC. In this context, the deregulated expression of miR-663, miR-708, and miR-375 was found to contribute to the emergence of NEPC by inducing neuroendocrine genes [161,162,163]. It was also demonstrated that miR-106a~363 cluster drives NEPC by pleiotropically regulating cardinal nodal poroteins such as Aurora Kinase A, N-Myc, E2F1, and STAT3 [162].

Other miRNAs have been reported to indirectly contribute to the emergence of treatment-induced neuroendocrine differentiation in PCa by modulating cellular pathways relevant to androgen independence and chemoresistance [164].

## 5. Conclusions

Experimental evidence reported in the review clearly indicates that specific miRNAs, whose expression is deregulated in PCa, not only play an important role in disease onset and progression but also represent molecular determinants of response/resistance to radio-, hormone-, and chemotherapy. Importantly, preclinical data form a solid foundation for promoting the use of miRNA-based strategies to modulate the therapeutic efficacy of ionizing radiation and anticancer drugs in PCa. In this context, specific miRNAs can be viewed as novel therapeutic targets or tools, and the modulation of their expression through the use miRNA inhibitors or mimics can be exploited to impair tumor growth and to enhance treatment response, mainly in highly aggressive subtypes of PCa, such as CRPC and NEPC.

However, there are several constraints towards translating miRNA-based strategies into human cancer therapy. A major constraint is related to the need for safe and efficient delivery systems able to guarantee an enhanced cell type-specific delivery. Different miRNA delivery systems have been proposed thus far, and some of them have demonstrated the ability to inhibit tumor growth in in vivo models of PCa [65,165,166,167,168]. Additional challenges in developing miRNA-based therapeutics are related to the need for a more detailed understanding of the functions exerted by specific miRNAs and the precise identification of their key targets relevant to PCa. Finally, the proper evaluation of safety profile of miRNA-based molecules as well as the improvement of currently available information concerning the pharmacokinetics of miRNA inhibitors and mimics will be instrumental for developing novel therapeutic strategies.

## Figures and Tables

**Figure 1 cancers-13-02380-f001:**
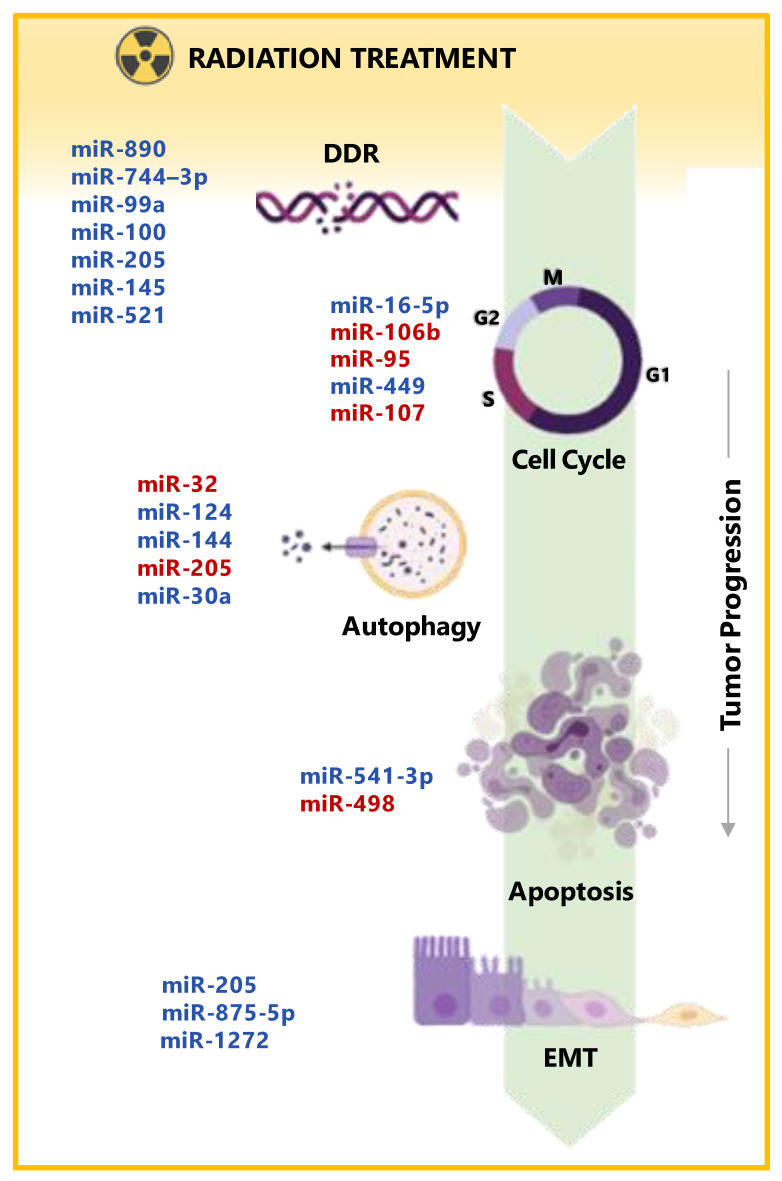
miRNAs involved in PCa response to radiotherapy. Schematic representation of the main biological mechanisms by which listed miRNAs concur to determine PCa response to radiation. miRNAs enhancing treatment response are highlighted in blue, while miRNAs inducing treatment resistance are in red. The reported mechanisms include DNA damage repair (DDR), cell cycle, autophagy, apoptosis, and epithelial-to-mesenchymal transition (EMT). Graphical elements were created with BioRender.com.

**Figure 2 cancers-13-02380-f002:**
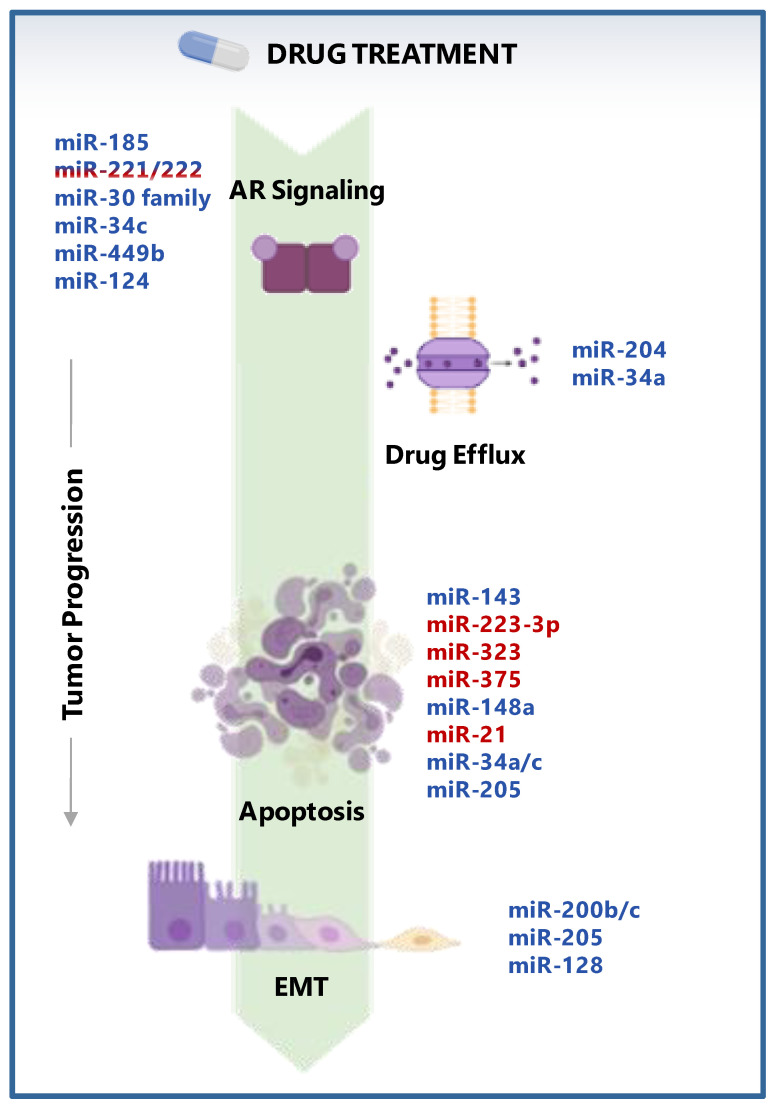
miRNAs involved in PCa drug treatment response. Schematic representation of the main biological mechanisms by which listed miRNAs concur to determine PCa response drug therapy. miRNAs enhancing treatment response are highlighted in blue, while miRNAs inducing treatment resistance are in red. The reported mechanisms include: androgen receptor (AR) signaling, drug efflux, apoptosis, and epithelial-to-mesenchymal transition (EMT). Graphical elements were created with BioRender.com.

**Table 1 cancers-13-02380-t001:** List of miRNAs implicated in PCa development.

miRNA	Expression in PCa	Target	Mechanism	References
miR-15/miR16	downregulated	Cyclin D1,WNT3A	Cell cycle	[67]
miR-205	downregulated	PKCε	Cell cycle	[68,69]
		PKCε, ZEB1	EMT	[70]
miR-34a	downregulated	SIRT1, Wnt, c-Myc, STMN	Apoptosis, cell cycle	[71,72,73,74]
miR-21	overexpressed	KLF5, PTEN	Apoptosis, cell proliferation, invasion and migration	[75,76]

**Table 2 cancers-13-02380-t002:** List of miRNAs involved in PCa response to radiotherapy.

miRNA	Target	Mechanism	Effect on Treatment	References
miR-106b	/	Cell cycle	resistance	[92]
miR-107	GRN	Cell cycle	sensitivity	[93]
miR-124	PIM1	Autophagy	sensitivity	[94]
miR-1272	HIP1	EMT	sensitivity	[95]
miR-144	PIM2	Autophagy	sensitivity	[94]
miR-145	SPOP	DDR	sensitivity	[96,97]
miR-16-5p	Cyclin D	Cell cycle	sensitivity	[98]
miR-205	PKCε	DDR	sensitivity	[69]
	TP53INP1	Autophagy	sensitivity	[99]
	PKCε, ZEB1	EMT	sensitivity	[69]
miR-30a	TP53INP1	Autophagy	sensitivity	[100]
miR-32	DAB2IP	Autophagy	resistance	[101]
miR-449	pRB/E2F1, c-Myc	Cell cycle	sensitivity	[102,103,104]
miR-498	PTEN	Apoptosis	resistance	[105]
miR-521	CSA	DDR	sensitivity	[106]
miR-541-3p	HSP27	Apoptosis	sensitivity	[107]
miR-744-3p	RAD23B	DDR	sensitivity	[108]
miR-875-5p	EGFR	EMT	sensitivity	[109]
miR-890	MAD2L2, WEE1, XPC	DDR	sensitivity	[108]
miR-95	SGPP1	Cell cycle	resistance	[110]
miR-99a/miR-100	SNF2H	DDR	sensitivity	[111]

**Table 3 cancers-13-02380-t003:** List of miRNAs involved in PCa response to ADT.

miRNA	Target	Mechanism	Effect on Treatment	References
miR-124	ARv7; ARv4; EZH2	AR signaling	sensitivity	[133]
miR-185	AR; BRD8 ISO2	AR signaling	sensitivity	[124,125]
miR-221/222	-	AR signaling	resistance in AR-sensitive cells; sensitivity in AR-independent cells	[127,128,129]
miR-30 family	AR; ARv7	AR signaling	sensitivity	[131]
miR-34c	ARv7; ARv4	AR signaling	sensitivity	[132]
miR-449b	ARv7; ARv4	AR signaling	sensitivity	[132]

**Table 4 cancers-13-02380-t004:** List of miRNAs involved in PCa response to chemotherapy.

miRNA	Target	Mechanism	Effect on Treatment	References
miR-128	ZEB1	EMT	sensitivity to cisplatin	[157]
miR-143	KRAS pathway	Apoptosis	sensitivity to docetaxel	[137]
miR-148a	MSK1	Apoptosis	sensitivity to taxanes	[141]
miR-200b/c	ZEB1; ZEB2	EMT	sensitivity to docetaxel	[154]
miR-204	ACSL4	Drug efflux	sensitivity to docetaxel	[158]
miR-205	Bcl-w; Bcl-2	Apoptois	sensitivity to docetaxel	[150,151]
	-	EMT	sensitivity to docetaxel	[154]
miR-21	PDCD4	Apoptosis	resistance to docetaxel	[144,160]
miR-223-3p	FOX3	Apoptosis	resistance to docetaxel	[138]
miR-323	p73	Apoptosis	resistance to docetaxel	[139]
miR-34a	SIRT1; Bcl-2	Apoptosis	sensitivity to docetaxel	[148]
	ACSL4; JAG1	Drug efflux	sensitivity to doxorubicin	[158,159]
miR-34c	-	Apoptosis	sensitivity to docetaxel	[149]
miR-375	YAP; SEC23A	Apoptosis	resistance to docetaxel	[140]

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
