# Peer review of "MicroRNAs as Epigenetic Determinants of Treatment Response and Potential Therapeutic Targets in Prostate Cancer"

_cancers, 2021, doi:10.3390/cancers13102380_

Round 1
Reviewer 1 Report
In this review, authors provide an overview on main PCa-related miRNAs and describe the biological mechanisms by which specific miRNAs concur to determine PCa response to radiation and drug therapy. Additionally, they discuss about the possibility of considering miRNAs such as novel therapeutic targets or tools based on the consequences of their expression modulation in PCa experimental models.
The manuscript reports a lot of interesting information but should be re-organized because it is very disordered.
For example, the par 1.3. “microRNAs in Cancer” is excessive and the part of introduction describing the therapies used in PC should be much clear and homogeneous with the following description of the par “1.2. Molecular contributors underlying PCa treatment response”; to reach this scope, in Introduction the authors should divide the therapies in: 1-Radiotherapy, 2- androgen deprivation therapy, 3- chemotherapy and indicate when these are used.
The role of AR in prostate cancer is very important, also in androgen resistant prostate cancer but the authors only briefly describe AR and its isoforms/mutants.
It is too hard read and remember all the miRNA described into the par “miRNA in prostate cancer”. The authors should describe, in this paragraph, the principal roles played by miRNA in PC without making a list of too much of miRNA. They should remember that the title of this manuscript is: “microRNAs as epigenetic determinants of treatment response and potential therapeutic targets in prostate cancer” and, for this reason, to extensively describe the miRNA in this field.
Furthermore, the authors neglect the androgen resistant PC forms in all the manuscript and especially in Conclusions.
Author Response
Reviewer 1:
Comments and Suggestions for Authors
In this review, authors provide an overview on main PCa-related miRNAs and describe the biological mechanisms by which specific miRNAs concur to determine PCa response to radiation and drug therapy. Additionally, they discuss about the possibility of considering miRNAs such as novel therapeutic targets or tools based on the consequences of their expression modulation in PCa experimental models.
The manuscript reports a lot of interesting information but should be re-organized because it is very disordered.
For example, the par 1.3. “microRNAs in Cancer” is excessive and the part of introduction describing the therapies used in PC should be much clear and homogeneous with the following description of the par “1.2. Molecular contributors underlying PCa treatment response”; to reach this scope, in Introduction the authors should divide the therapies in: 1-Radiotherapy, 2- androgen deprivation therapy, 3- chemotherapy and indicate when these are used.
Answer: We thank the Reviewer for the comments and the interest in our review article. Concerning the section “microRNAs in Cancer”, we briefly described the involvement of miRNAs in the regulation of multiple cell phenomena related to cancer development and progression. This chapter has been written in order to provide a concise overview of the role of miRNA in cancer, useful for outlining the topic and extend the readership also to people not specialized in the field. As suggested by the Reviewer, we divided the Introduction - chapter 1.1. “Prostate cancer management”- according to the different therapies, following the scheme used for the chapter 1.2 “Molecular contributors underlying PCa treatment response” (page 2, line 22 and 46; page 3, line 19).
The role of AR in prostate cancer is very important, also in androgen resistant prostate cancer but the authors only briefly describe AR and its isoforms/mutants.
Answer: To simultaneously answer the requests of Reviewers n. 1 and n. 2, an additional chapter describing the more relevant deregulated pathways involved in prostate cancer growth and progression has been added. In this chapter entitled “1.2 Key pathways involved in PCa growth and disease progression”, the role of AR in prostate cancer is stated (page 3, lines 45-56; page 4 lines 1-40).
It is too hard read and remember all the miRNA described into the par “miRNA in prostate cancer”. The authors should describe, in this paragraph, the principal roles played by miRNA in PC without making a list of too much of miRNA. They should remember that the title of this manuscript is: “microRNAs as epigenetic determinants of treatment response and potential therapeutic targets in prostate cancer” and, for this reason, to extensively describe the miRNA in this field.
Answer: When we wrote the manuscript, our intention was to provide a brief overview concerning the most relevant miRNAs involved in prostate cancer. Indeed, among the huge amount of miRNAs, whose expression/function was reported as relevant for the disease, we focused on four miRNAs, for which solid data is available. As suggested by the Reviewer, in order to make clearer the role of the selected miRNAs, we summarized the main findings in an additional table (page 9,line 23).
In addition, a few sentences have been added to the manuscript concerning the feedback networks between miRNAs and epigenetic modifications in cancer (page 6, lines 44-50).
Furthermore, the authors neglect the androgen resistant PC forms in all the manuscript and especially in Conclusions.
Answer: As suggested by the Reviewer, we provided additional comments regarding the androgen resistant prostate cancer forms. Specifically, we added i) a few sentences dealing with the emergence of AR independent neuroendocrine PCa (NEPC) in the Introduction (page 3, lines 34-39), and ii) a short chapter focused on the main miRNAs involved in NEPC development (page 21, lines 1-12). In addition, we mentioned in the Conclusion the relevance of exploring novel miRNA-based therapeutic tools in order to enhance treatment response in highly aggressive subtypes of PCa such as castration resistant prostate cancer and NEPC (page 21, lines 22-23).
Reviewer 2 Report
In the review article, the researchers described the role of microRNAs as epigenetic determinants of treatment response and potential therapeutic targets in prostate cancer. The manuscript is interesting. Nevertheless, it should be improved.
- The authors should add a chapter describing the most significant pathways in prostate cancer growth and progression before describing the role of the selected factors in therapies.
- The role of microRNAs in the selected processed and therapies affecting prostate cancer growth should be summarized in more tables or diagrams. The authors presented only two tables showing miRNAs important in PC response to radiotherapy and general drug response, but the manuscript described more processes and therapies. Presentation of data in diagrams of tables will make the article easier to follow.
- Misspellings and double spaces should be corrected.
Author Response
Reviewer 2:
Comments and Suggestions for Authors
In the review article, the researchers described the role of microRNAs as epigenetic determinants of treatment response and potential therapeutic targets in prostate cancer. The manuscript is interesting. Nevertheless, it should be improved.
The authors should add a chapter describing the most significant pathways in prostate cancer growth and progression before describing the role of the selected factors in therapies.
Answer: We thank the Reviewer for the comments and the interest in our review article. In order to follow the Reviewer suggestion, a chapter describing the most relevant pathways involved in prostate cancer growth and progression has been added, particularly focusing on such pathways that play a crucial role in response to therapy (page 3, lines 45-53; page 4, lines 1-40).
The role of microRNAs in the selected processed and therapies affecting prostate cancer growth should be summarized in more tables or diagrams. The authors presented only two tables showing miRNAs important in PC response to radiotherapy and general drug response, but the manuscript described more processes and therapies. Presentation of data in diagrams of tables will make the article easier to follow.
Answer: Following the Reviewer suggestion, the number of Tables and Figures has been increased (new Fig. 1, page 15; Fig. 2 page 20; Table 1, page 9; Table 2, pages 15-16; Table 3, pages 17-18; Table 4, pages 20-21).
Misspellings and double spaces should be corrected.
Answer: Misspellings and double spaces have been corrected in the revised manuscript.
Reviewer 3 Report
The authors in this review "microRNAs as epigenetic determinants of treatment response and potential therapeutic targets in prostate cancer" describe the topic in an exhaustive way. Firstly, they frame the PCa issue well in both pathological and therapeutic aspects, addressing in detail the focus of the review, that is, the role of microRNAs in the multiple facets of PCa and in response to different therapies. The figures and second tables are also complete and very clear.
Author Response
Reviewer 3:
Comments and Suggestions for Authors
The authors in this review "microRNAs as epigenetic determinants of treatment response and potential therapeutic targets in prostate cancer" describe the topic in an exhaustive way. Firstly, they frame the PCa issue well in both pathological and therapeutic aspects, addressing in detail the focus of the review, that is, the role of microRNAs in the multiple facets of PCa and in response to different therapies. The figures and second tables are also complete and very clear.
Answer: We thank the Reviewer for his/her positive comment regarding our review manuscript.
Round 2
Reviewer 1 Report
The authors satisfactorily reply to all my comments.